# Visual-Interactive Neural Machine Translation

Tanja Munz*
University of Stuttgart

Dirk Väth†
University of Stuttgart

Paul Kuznecov
University of Stuttgart

Ngoc Thang Vu‡
University of Stuttgart

Daniel Weiskopf§
University of Stuttgart

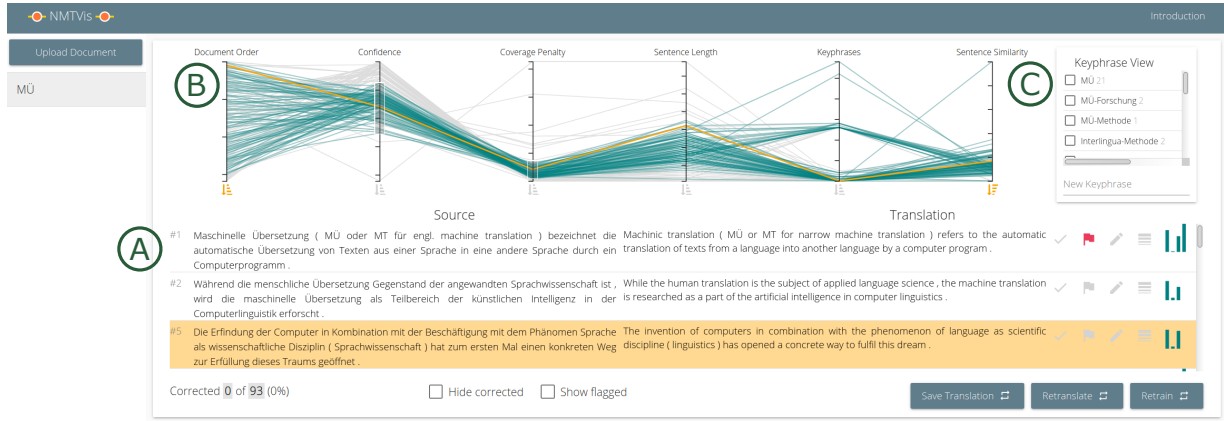

Figure 1: The main view of our neural machine translation (NMT) system: (A) Document View with sentences of the document for the current filtering settings, (B) Metrics View with sentences of the filtering result highlighted, and (C) Keyphrase View with a set of rare words that may be mistranslated. The Document View initially contains all sentences automatically translated with the NMT model. After filtering with the Metrics View and Keyphrase View, a smaller selection of sentences is shown. Each entry in the Document View provides information about metrics, the correction state, and functionality for modification (on the right side next to each sentence). The Metrics View represents each sentence as one path and shows values for different metrics (e.g., correlation, coverage penalty, sentence length). Green paths correspond to sentences of the current filtering. One sentence is highlighted (yellow) in both the Metrics View and the Document View.

## ABSTRACT

We introduce a novel visual analytics approach for analyzing, understanding, and correcting neural machine translation. Our system supports users in automatically translating documents using neural machine translation and identifying and correcting possible erroneous translations. User corrections can then be used to fine-tune the neural machine translation model and automatically improve the whole document. While translation results of neural machine translation can be impressive, there are still many challenges such as over- and under-translation, domain-specific terminology, and handling long sentences, making it necessary for users to verify translation results; our system aims at supporting users in this task. Our visual analytics approach combines several visualization techniques in an interactive system. A parallel coordinates plot with multiple metrics related to translation quality can be used to find, filter, and select translations that might contain errors. An interactive beam search visualization and graph visualization for attention weights can be used for post-editing and understanding machine-generated translations. The machine translation model is updated from user corrections to improve the translation quality of the whole document. We designed our approach for an LSTM-based translation model and extended it to also include the Transformer architecture. We show for representative examples possible mistranslations and how to use our system to deal with them. A user study revealed that many

*tanja.munz@visus.uni-stuttgart.de

†dirk.vaeth@ims.uni-stuttgart.de

‡thangvu@ims.uni-stuttgart.de

§daniel.weiskopf@visus.uni-stuttgart.de

participants favor such a system over manual text-based translation, especially for translating large documents.

**Index Terms:** Human-centered computing—Visualization—Visualization application domains—Visual analytics; Human-centered computing—Visualization—Visualization systems and tools; Computing methodologies—Artificial intelligence—Natural language processing—Machine translation

## 1 INTRODUCTION

Machine learning and especially deep learning are popular and rapidly growing fields in many research areas. The results created with machine learning models are often impressive but sometimes still problematic. Currently, much research is performed to better understand, explain, and interact with these models. In this context, visualization and visual analytics methods are suitable and more and more often used to explore different aspects of these models. Available techniques for visual analytics in deep learning were examined by Hohman et al. [19]. While there is a large amount of work available for explainability in computer vision, less work exists for machine translation.

As it becomes increasingly important to communicate in different languages, and since information should be available for a huge range of people from different countries, many texts have to be translated. Doing this manually takes much effort. Nowadays, online translation systems like Google Translate [13] or DeepL [10] support humans in translating texts. However, the translations generated that way are often not as expected or like someone familiar with both languages might translate them. It may also not express someone's translation style or use the correct terminology of a specific domain or for some occasion. Often, more background knowledge about the text is required to translate documents appropriately.

With the introduction of deep learning methods, the translation quality of machine translation models has improved considerably

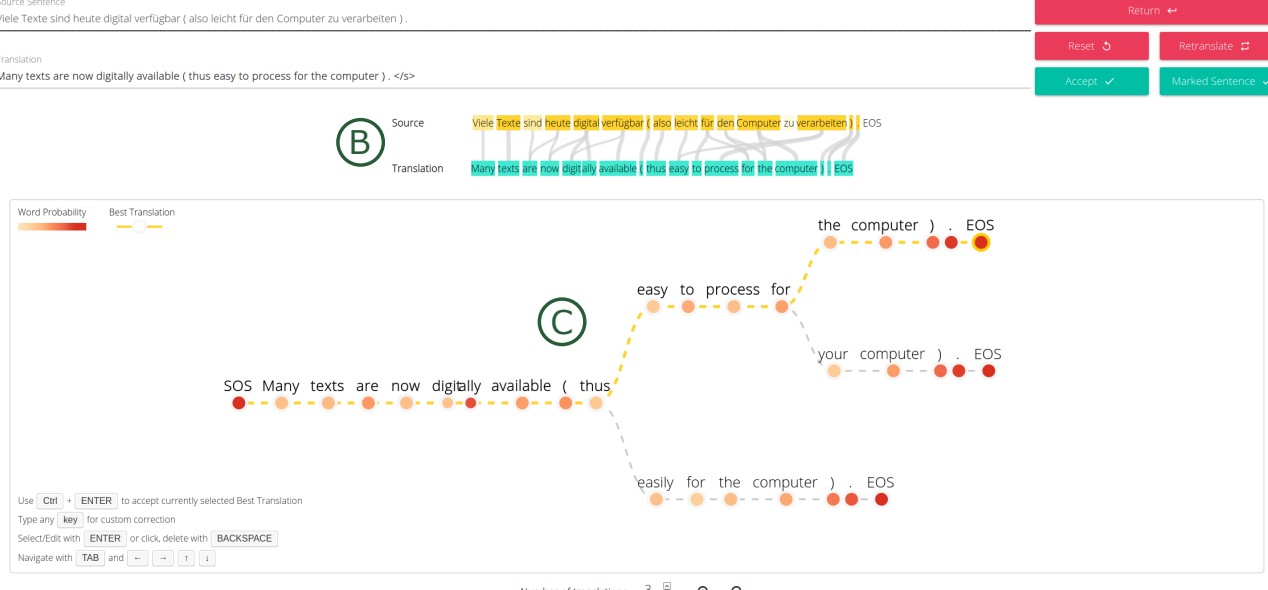

Figure 2: The detailed view for a selected sentence consists of the Sentence View (A), the Attention View (B), and the Beam Search View (C). The Sentence View allows text-based modifications of the translation. The Attention View shows the attention weights (represented by the lines connecting source words with their translation) for the translation. The Beam Search View provides an interactive visualization that shows different translation possibilities and allows exploration and correction of the translation. All three areas are linked.

in the last years. However, there are still difficulties that need to be addressed. Common problems of neural machine translation (NMT) models are, for instance, over- and under-translation [41] when words are translated repeatedly or not at all. Handling rare words [23], which might be available in specific documents, and long sentences, are also issues. Domain adaption [23] is another challenge; especially documents from specific domains such as medicine, law, or science require high-quality translations [7]. As many NMT models are trained on general data sets, their translation performance is worse for domain-specific texts.

If high-quality translations for large texts are required, it is insufficient to use machine translation models alone. These models are computationally efficient and able to translate large documents with low time effort, but they may create erroneous or inappropriate translations. Humans are very slow compared to these models, but they can detect and correct mistranslations when familiar with the languages and the domain terminology. In a visual analytics system, both of these capabilities can be combined. Such a system should provide the translations from an NMT model and possibilities for users to visually explore translation results to find mistranslated sentences, correct them, and steer the machine learning model.

We have developed a visual analytics approach to reach the goals outlined above. First, our system performs automatic translation of a whole, possibly large, document and shows the result in the Document View (Figure 1). Users can then explore and modify the document on different views [34] (Figure 2) to improve translations and use these corrections to fine-tune the NMT model. We support different NMT architectures and use both an LSTM-based and a Transformer architecture.

So far, visual analytics systems for deep learning were mostly available for computer vision, some text-related areas, focusing on smaller parts of machine translation [26, 33] or intended for domain experts to gain insight into the models or to debug them [38, 39]. This work contributes to visualization research by introducing the application domain of NMT using a user-oriented visual analytics approach. In our system, we employ different visualization techniques

adapted for usage with NMT. Our parallel coordinates plot (Figure 1 (B)) supports the visualization of different metrics related to text quality. The interaction techniques in our graph-based visualization for attention (Figure 2 (B)) and tree-based visualization for beam search (Figure 2 (C)) are specifically designed for text exploration and modification. They have a strong coupling to the underlying model. Furthermore, our system has a fast feedback loop and allows interaction in real-time. We demonstrate our system's features in a video and provide the source code[1] [29] of our system along with the trained models we used in our case study [30].

## 2  RELATED WORK

This section first discusses visualization, visual analytics, and interaction approaches for language translation in general and then visual analytics of deep learning for text. Afterward, we provide an overview of work that combines both areas in the context of NMT.

Many visualization techniques and visual analytics systems exist for text; see Kucher and Kerren [24] for an overview. However, there is little work on exploring and modifying translation results. An interactive system to explore and correct translations was introduced by Albrecht et al. [1]. While the translation was created by machine translation, their system did not use deep learning. Lattice structures with uncertainty visualization were employed by Collins et al. [9] in the context of machine translation. They created a lattice structure from beam search where the path for the best translation result is highlighted and can be corrected. We also use visualization for beam search, but ours is based on a tree structure. Without the use of visualization, Green et al. [15] follow a similar approach to ours to let users correct machine-translated sentences providing suggestions. They discussed that post-editing of mistranslated sentences reduces time and creates results with better quality [16, 17].

Recently, much research was done to visualize deep learning models to understand them better. Multiple surveys [6, 12, 19, 27, 50] are available that provide summaries of existing visual analytics sys-

---

[1]https://github.com/MunzT/NMTVis

tems. It is noticeable that not much work exists related to text-based domains. One of the few examples is RNN-Vis [28], a visual analytics system designed to understand and compare models for natural language processing by considering hidden state units. Karpathy et al. [21] explore the prediction of Long Short-Term Memory (LSTM) models by visualizing activations on text. Heatmaps are used by Hermann et al. [18] in order to visualize attention for machine-reading tasks. To explore the training process and to better understand how the network is learning, RNNbow [4] can be used to visualize the gradient flow during backpropagation training in Recurrent Neural Networks (RNNs).

While the previous systems support the analysis of deep learning models for text domains in general, approaches exist to specifically explore and understand NMT. The first who introduced visualizations for attention were Bahdanau et al. [2]; they showed the contribution of source words to translated words within a sentence, using an attention weight matrix. Later, Rikters et al. [33] introduced multiple ways to visualize attention and implemented exploration of a whole document. They visualize attention weights with a matrix and a graph-based visualization connecting source words and translated words by lines whose thickness represents the attention weight. Bar charts give an overview for a whole document for multiple attention-based metrics that are supposed to correlate with the translation quality. Interactive ordering of these metrics and sentence selection is possible. However, it is difficult for large documents to compare the different metrics as each bar chart is horizontally too large to be entirely shown on a display. The only connection between different bar charts is that the bars for the currently selected sentence are highlighted. Our system also uses such a metrics approach, but instead of relying on bar charts, a parallel coordinates plot was chosen for better scalability, interaction, and filtering.

An interactive visualization approach for beam search is provided by Lee et al. [26]. The interaction techniques supported by their tree structure are quite limited. It is possible to expand the structure and to change attention weights. However, it is not possible to add unknown words, and no sub-word units are considered. Furthermore, the exploration is limited to single sentences instead of a whole document.

With LSTMVis, Strobelt et al. [39] introduced a system to explore LSTM networks by showing hidden state dynamics. Among other application areas, their approach is also suitable for NMT. While our approach is rather intended for end-users, LSTMVis has the goal of debugging models by researchers and machine learning developers. With Seq2Seq-Vis, Strobelt et al. [38] present a system that uses an attention view similar to ours, and they also provide an interactive beam search visualization. However, their system is designed to translate single sentences, and no model adaption is possible for improved translation quality. Their system aims at debugging and gaining insight into the models.

Since there are different architectures available for generating translations [49], specific visualization approaches may be required. Often, LSTM-based architectures are used. Recently, the Transformer architecture [42] has gained popularity; Vig [43, 44] visually explores their self-attention layers and Rikters et al. [32] extended their previous approach for debugging documents to Transformer-based systems.

All these systems provide different, possibly interactive, visualizations. However, their goal is rather to debug NMT models instead of supporting users in translating entire documents, or they are limited to small aspects of the model. Additionally, they are usually designed for one specific translation model. None of these approaches provide extended interaction techniques for beam search or interactive approaches to iteratively improve the translation quality of a whole document. We did some preliminary work for this project in a master's thesis [25].

## 3 VISUAL ANALYTICS APPROACH

Our visual analytics approach allows the automatic translation, exploration, and correction of documents. Its components can be split into multiple parts. First, a document is automatically translated from one language into another one, then mistranslated sentences in the document are identified by users, and individual sentences can be explored and corrected. Finally, the model can be fine-tuned and the document retranslated.

Our approach has a strong link to machine data processing and follows the visual analytics process presented by Keim et al. [22]. We use visualizations for different aspects of NMT models, and users can interact with the provided information.

### 3.1 Requirements

For the development of our system, we followed the nested model by Munzner [31]. The main focus was on the outer parts of the model, including identifying domain issues, feature implementation design, and visualization and interaction implementation. Additionally, we used a similar process as Sedlmair et al. [35], especially focusing on the core phases. Design decisions were made in close cooperation with deep learning and NMT experts, who are also co-authors of this paper. The visual analytics system was implemented in a formative process that included these experts. Our system went through an iterative development that included multiple meetings with our domain experts. Together, we identified the requirements listed in Table 1. After implementing the basic prototype of the system, we demonstrated it to further domain experts. At a later stage, we performed a small user study with experts for visualization and machine translation. For our current prototype, we added recommended functionality from these experts.

### 3.2 Neural Machine Translation

The goal of machine translation is to translate a sequence of words from a source language into a sequence of words in a target language. Different approaches exist to achieve this goal [40, 49].

Usually, neural networks for machine translation are based on an encoder-decoder architecture. The encoder is responsible for transforming the source sequence into a fixed-length representation known as a context vector. Based on the context vector, the decoder generates an output sequence where each element is then used to generate a probability distribution over the target vocabulary. These probabilities are then used to determine the target sequence; a common method to achieve this uses beam search decoding [14].

Although different NMT models vary in their architecture, the previously described encoder-decoder design should apply to a wide range of architectures and new approaches that may be developed in the future (R6). In this work, we explored an LSTM architecture with attention and extended our approach to include the Transformer architecture, thus verifying its ability to generalize.

One of the first neural network architectures for machine translation consists of two RNNs with LSTM units [5]. To handle long sentences, the attention mechanism for NMT [2] was introduced. It allows sequence-to-sequence models to pay attention to different sections of the input sequence while predicting the next item of the output sequence by providing the decoder access to the encoder's weighted hidden states. During decoding, the hidden states of the encoder together with the hidden state of the decoder for the current step are used to compute the attention scores. Finally, the context vector for the current step is computed as a sum of the encoder hidden states, weighted by the attention scores. The attention weights can be easily visualized and used to explain why a neural network model predicted a certain output. Furthermore, the attention weights can be seen as a soft alignment between source and target sequences. For each translated word, the weight distribution over the source sequence indicates which source words were most relevant for predicting that target word. The Transformer architecture was recently

Table 1: Requirements for our visual analytics system and their implementations in our approach.

| R1 | **Automatic translation** – A document is translated automatically by an NMT model. |
|----|------|
| R2 | **Overview** – The user can see the whole document as a list of all source sentences and their translations (Figure 1 (A)). Additionally, an overview of the translation quality is provided in the Metrics View that reveals statistics about different metrics encoded as a parallel coordinates plot (Figure 1 (B)) showing an overall quality distribution. |
| R3 | **Find, filter, and select relevant sentences** – Interaction in the parallel coordinates allows filtering according to different metrics and selecting specific sentences. It is also possible to select one sentence and order the other sentences of the document by similarity to verify for similar sentences if they contain similar errors. Additionally, our Keyphrase View (Figure 1 (C)) supports selecting sentences containing specific keywords that might be domain-specific and rarely used in general documents. |
| R4 | **Visualize and modify sentences** – For each sentence, a beam search and attention visualization (Figure 2) can be used to interactively explore and adapt the translation result in order to correct erroneous sentences and explore how a translation failed. It is also possible to explore alternative translations. |
| R5 | **Update model and translation** – The model can be fine-tuned using the user inputs from translation corrections; this is especially useful for domain adaption. Afterward, the document is retranslated with the updated model in order to improve the translation result (the result is visualized similar to Figure 9). |
| R6 | **Generalizability and extensibility** – While we initially designed our visualization system for one translation model, we soon noticed that our approach should handle data from other translation models as well. Therefore, our approach should be easily adaptable for new models to cope with the dynamic development of new deep learning architectures. Our general translation and correction process is held quite agnostic to be applied on a variety of models. Model-specific visualizations may have limitations and need to be adapted or exchanged when using a different translation architecture. |
| R7 | **Target group** – The target group for our system should be quite broad and include professional translators or students who need to translate documents. However, it should also be able to be used by other people interested in correcting and possibly better understanding the results of automated translation. |

introduced by Vaswani et al. [42] and gained much popularity. It uses a more complex attention mechanism with multi-head attention layers; especially, self-attentions play an important role in the translation process. We verify its applicability to our approach and visualize only the part of the attention information that showed an alignment between source and target sentences comparable to the LSTM model.

### 3.3 Exploration of Documents

After uploading a document to our system, it is translated by an NMT model (R1). The main view of our approach then shows information about the whole document (R2). This includes a list of all sentences in the Document View (Figure 1 (A)) and an overview of the translation quality in the Metrics View (Figure 1 (B)). Using the Metrics View and Keyphrase View (Figure 1 (C)), sentences can be filtered to detect possible mistranslated sentences that can be flagged by the user (R3). Once a mistranslated sentence is found, it is also possible to filter for sentences containing similar errors (R3).

#### Metrics View

In the *Metrics View*, a parallel coordinates plot (Figure 1 (B)) is used to detect possible mistranslated sentences by filtering sentences according to different metrics (R3). For instance, it is possible to find sentences that have low translation confidence.

Multiple metrics exist that are relevant to identify translations with low quality; we use the following metrics in our approach:

- **Confidence:** A metric that considers attention distribution for input and output tokens; it was suggested by Rikters et al. [33]. Here, a higher value is usually better.
- **Coverage penalty:** This metric by Wu et al. [48] can be used to detect sentences where words did not get enough attention. Here, a lower value is usually better.
- **Sentence length:** The sentence length (the number of words in a source sentence) can be used to filter very short or long sentences. For example, long sentences might be more likely to contain errors.
- **Keyphrases:** This metric can be used to filter for sentences containing domain-specific words. As these words are rare in the training data, the initial translation of sentences containing them is likely erroneous. The values used for this metric are the number of occurrences of keyphrases in a sentence weighted by the frequency of the keyphrases in the whole document.
- **Sentence similarity:** Optionally, for a given sentence, the similarity to all other sentences can be determined using cosine similarity. This helps to find sentences with similar errors to a detected mistranslated sentence.
- **Document index:** The document index allows the user to sort sentences according to their original order in the document, which can be especially important for correcting translations where the context of sentences is relevant. Furthermore, this metric might also show trends like consecutive sentences with low confidence.

In contrast to Rikters et al. [33], who use bar charts to visualize different metrics, we chose a parallel coordinates plot [20]. Each sentence can be mapped to one line in such a plot, and different metrics can be easily compared. These plots are useful for an overview of different metrics and to detect outliers and trends. Interactions with the metrics, such as highlighting lines or choosing filtering ranges, are supported. It can be expected that sentences filtered for both, low confidence and high coverage penalty, are more likely to be poorly translated than sentences falling into only one of these categories.

#### Keyphrase View

It is possible to search for sentences according to keyphrases by selecting them in the *Keyphrase View* (Figure 1 (C)) (R3). This can be visualized as shown in Figure 4. Keyphrases are domain-specific words and were not often included in the training data used for our model since we trained our model on general data. As the model has not enough knowledge on how to deal with these words, it is important to verify if the respective sentences were translated correctly. In addition to automatically determined keyphrases, users can manually specify further keyphrases for sentence filtering.

#### Document View

A list of all the source sentences in a document and a list of their translations are shown in the *Document View* (Figure 1 (A)) (R2). Each entry in this list can be marked as correct or flagged (Figure 4) for later correction. A small histogram shows an overview of the previously mentioned metrics. If a sentence is modified, either through user-correction or retranslation by the fine-tuned model, changes in the sentences are highlighted (Figure 9). Both the Metrics View and the Keyphrase View are connected via brushing and linking [45] to allow filtering for sentences that are likely to be mistranslated and should be examined and possibly corrected. Additionally, sentences can be sorted into a list by similarity to a user-selected reference sentence. In this list, sentences can be selected for further exploration and correction in more detailed sentence-based views.

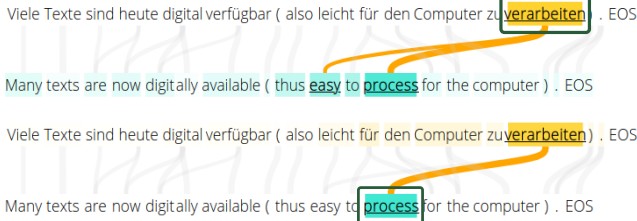

Figure 3: Attention visualization: (top) when hovering a source word (here: 'verarbeiten') translated words influenced by the source are highlighted and (bottom) when hovering a translated word (here: 'process') source words that influence the translation are highlighted according to attention weights.

## 3.4 Exploration and Correction of Sentences

After filtering and selection, a sentence can be further analyzed with the Sentence, Attention, and Beam Search Views (Figure 2) and subsequently corrected (R4). These views are shown simultaneously to allow interactive exploration and modification of translations.

Note, on the sentence level, we use subword units to handle the problem of rare words, which often occurs in domain-specific documents, and to avoid unknown words. We use the Byte Pair Encoding (BPE) method proposed by Sennrich et al. [37] for compressing text by recursively joining frequent pairs of characters into new subwords. This means, instead of using whole words to build the source and target vocabulary, words are split into subword units consisting of possibly multiple characters. This method reduces model size, complexity, and training time. Additionally, the model can handle unknown words by splitting them into their subword units. As these subword units are known beforehand, they do not require the introduction of an "unknown" token for translation. Thus, we can adapt the NMT model to any new domain, including those with vocabulary not seen at training time.

### Sentence View

Similar to common translation systems, the *Sentence View* (Figure 2 (A)) shows the source sentence and the current translation. It is possible to manually modify the translation, which in turn updates the content in the other sentence-based views. After adding a new word in the text area, the translation with the highest score is used for the remainder of the sentence. This supports a quick text-based modification of a translation without explicit use of visualizations. Currently, changing the translation updates the whole sentence after the modified word. Therefore, we do not support deleting or changing words while maintaining the remainder of a sentence.

### Attention View

The *Attention View* depends on the underlying NMT model. It is intended to visualize the relationship between words of the source sentence and the current translation as a weighted graph (Figure 2 (B)); such a technique was also used by Strobelt et al. [38]. Both source and translated words are represented by nodes; links between such words show the attention weights encoded by the thickness of the connecting lines (we use a predefined threshold to hide lines for very low attention). These weights correlate with the importance of source words for the translated words. Hovering over a source word highlights connecting lines to translated words starting at this word. In addition, the translated words are highlighted by transparency according to the attention weights (Figure 3 top). While this shows how a source word contributes to the translation, it is also possible to show for translated words how source words contribute to the translation (Figure 3 bottom). This interactive visualization supports users in understanding how translations are generated from the

source sentence words. On the one hand, such a visualization helps gain insight into the NMT model, and, on the other hand, it helps detect issues in generated translations. The links between source sentence and translation can be explored to identify anomalies such as under- or over-translation. Missing attention weights can be an indication for under-translation and links to multiple translated words for over-translation. In our case study in Section 4, examples of these cases are presented. While this technique specifically employs information of the attention-based LSTM model, we use it in an adapted form for the Transformer architecture (see Section 4.4). A visualization more tailored to Transformers, also including self-attention and attention scores from multiple decoder layers, could provide additional information. Further models may need different visualizations for a generalized use of our approach, employing model-specific information.

### Beam Search View

While the Attention View can be used to identify positions with mistranslations, the *Beam Search View* supports users in interactively modifying and correcting translations. The Beam Search View visualizes multiple translations created by the beam search decoding as a hierarchical structure (see Figure 2 (C)). This interactive visualization can be used for post-editing the translations.

The simplest way of predicting a target sequence is greedy decoding. In each time step, the token with the highest output probability is chosen as the next predicted token and fed to the decoder in the following step. This is an efficient and simple way to generate an output sequence. However, another translation may be better overall, despite having lower probabilities for the first words. Beam search decoding [14] is a tradeoff between exhaustive search and greedy decoding, often used for generating the final translation. At each time step, a predefined number ($k$) of hypotheses is considered. For each hypothesis, the NMT model outputs a probability distribution over the target vocabulary for the next token. These hypotheses are sorted by the probability of the final token, and up to $k$ hypotheses remain in the beam. Hypotheses that end with the End-of-Sequence (EOS) token are selected to build the result set. Once $k$ hypotheses stay in the result set, the algorithm stops, and the final hypotheses are ranked according to a score function that depends on attention weights and sentence length.

For visualization, we use a similar approach as Strobelt et al. [38], and Lee et al. [26]: a tree structure reflects the inherently hierarchical nature of the beam search decoding. This way, translation hypotheses starting with the same prefixes are merged into one branch of this hierarchical structure. The root node of each translation is associated with a Start-of-Sequence (SOS) token and all leaf nodes with an End-of-Sequence (EOS) token. Compared to the visualization of a list of different suggested translations, showing a tree is more compact, and it is easier to recognize where commonalities of different translation variants lie.

Each term of the translation is visualized by a circle that represents the node and a corresponding label. The color of a circle is mapped to the word's output probability. This supports users in identifying areas with a lower probability that might require further exploration. It can be seen as uncertainty for the prediction of words. In our visualization, we differentiate between nodes that represent subwords and whole words. Continuous lines connect subwords and nodes are placed closer together to form a unit. In contrast, the connections to whole words are represented by dashed lines.

The beam search visualization can be used to navigate within a translation and edit it (Figures 7 and 8). The interaction can be realized with mouse or keyboard input; the latter is more efficient for fast post-editing. The view supports standard panning-and-zooming techniques that are especially needed to explore long sentences as they do not fit common displays. For navigation within the tree, arrow keys can be used to move through a sentence, or nodes can

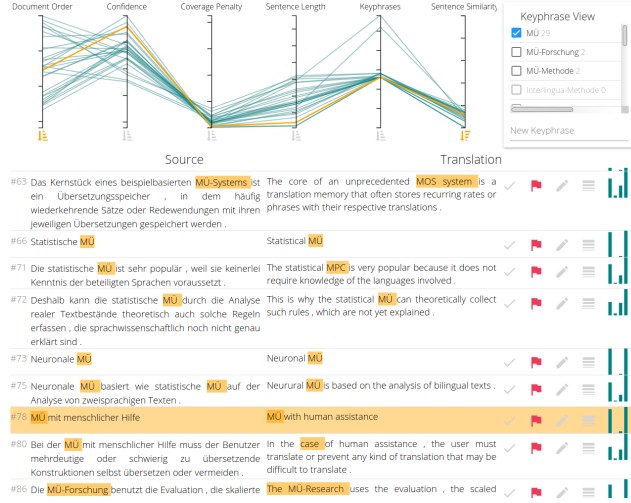

Figure 4: Main view of the system: the Document View shows some flagged sentences for correction. Additionally, the keyphrase filter (top right) is active: all sentences containing the keyphrase 'MÜ' are shown in the Metrics and Document Views. It is visible that 'MÜ' is never correctly translated to 'MT'.

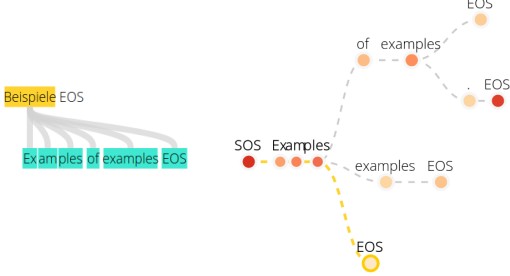

Figure 5: Example of over-translation: 'Examples' is placed twice as translation for the German word 'Beispiele'. The Beam Search View (right) shows possible alternative translations. However, only increasing the beam size to four shows the translation we would have expected.

be selected by mouse cursor. If the translation of the current node's child node is not satisfying, the node can be expanded to show suggestions for correction. If the user selects a suggested word, the beam search runs with a lexical prefix constraint, and the tree structure gets updated. If the suggested words are not suitable, a custom correction can be performed by typing an arbitrary word that better fits. The number of suggested translations is initially set to three and can be increased by adapting the beam size. Increasing this value may create better translations and provides more alternative translations. However, the higher the value is, the more information has to be shown in the visualization. By hovering and selecting elements in this view, corresponding elements of the Attention View and Sentence View are shown for reference.

## 3.5 Model Fine-tuning and Retranslation

After correcting the translation of multiple sentences, the user corrections can be used to fine-tune the NMT model and automatically improve the translation of the not yet verified sentences (R5). This approach can be applied repeatedly to improve the document's translation quality, especially for domain-specific texts.

Documents often belong to a specific domain, such as legal, medical, or scientific. Each such domain uses a specific vocabulary, and the same word can even mean different things in different domains. Therefore, the capability of NMT models to handle different types of domains is very important. Domain adaptation means that NMT models trained on general training data (out-of-domain) can adapt to domain-specific documents (in-domain). This is useful because there is a large amount of general training data, but domain-specific data is rare. Since NMT models require a very large amount of training data to achieve good translation quality, the out-of-domain data can be used to train a baseline model. The model can be fine-tuned using in-domain data (R5), which usually contains a small number of sentences: we use the user-corrected sentences in our system. This mitigates the problem of training an NMT model where not much data exists for a particular domain. In our approach, we continue training for the in-domain data in a reduced way by freezing certain model weights (for the LSTM-based model, both decoder and the LSTM layers of the encoder are trained; for the Transformer, only the decoder is trained).

## 4 CASE STUDY

As a typical use case, we take the German Wikipedia article for *machine translation* (*Maschinelle Übersetzung*) [47] as a document for translation into English. In the following, we show how to use our system to improve the translation quality of the document. Please see our accompanying video for a demonstration with the Transformer model. The examples in the following were created with both the LSTM and Transformer models. We trained our models on a general data set: the German-to-English data set from the 2016 ACL Conference on Machine Translation (WMT'16) [3] shared news translation task. This is a popular data set for NMT, used, for instance, by Denkowski and Neubig [11] and Sennrich et al. [36].

### 4.1 Exploration of Documents

After uploading a document (R1), we have a look at the parallel coordinates plot (R2) for our initial translations and the list of keyphrases in order to detect possible mistranslations (R3). In the Keyphrase View, we notice the domain-specific term 'MÜ' occurring very often. This term is the German abbreviation for 'machine translation' and should therefore be translated as 'MT'. However, none of the translations use the correct term (Figure 4). Additionally, one could select and verify sentences with low confidence or with a high coverage penalty. Here, we especially notice the under-translation of some sentences. After verifying a translation in the Document View, users can decide if they are correct (R2). If the users do not agree with the translation, they can set a flag (Figure 4) to modify the translation later or switch to the sentence-based views to correct it (R4).

### 4.2 Exploration and Correction of Sentences

After setting flags for multiple sentences (Figure 4), or the decision to explore or modify a sentence, a more detailed view for each sentence can be shown to explore and improve their translations interactively (Figure 2) (R4).

**Over-translation** is a common issue of NMT [23]. In the Attention View, it is possible to see what went wrong by identifying where the attention weights connect the source and destination words.

For both models, we notice some cases for very short sentences. Figure 5 shows for the German heading 'Beispiele' (en: 'Examples'), a translation that uses the translated word multiple times. Also, the suggested alternatives use this term more than once. Only after increasing the beam size to four, the correct translation is visible, which can then be selected as the correction.

More often, only parts of a sentence are translated, and important words are not considered in our document. Such **under-translation** is shown in Figure 6. In the first example, only the beginning of the sentence is translated, and it is visible that the remaining nodes have almost zero attention. In the second example, the German term 'zweisprachigen' (en: 'bilingual') is skipped in the translation.

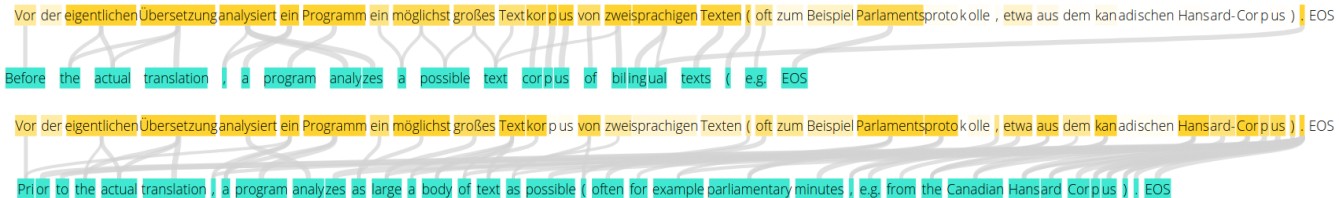

Figure 6: Example of under-translation shown in the Attention View: (top) For the LSTM model, the end of the sentence is not translated; attention weights are very low for this part of the sentence. (Bottom) For the Transformer architecture, the term 'zweisprachigen' (en: 'bilingual') is not translated; attention weights are very low for this term.

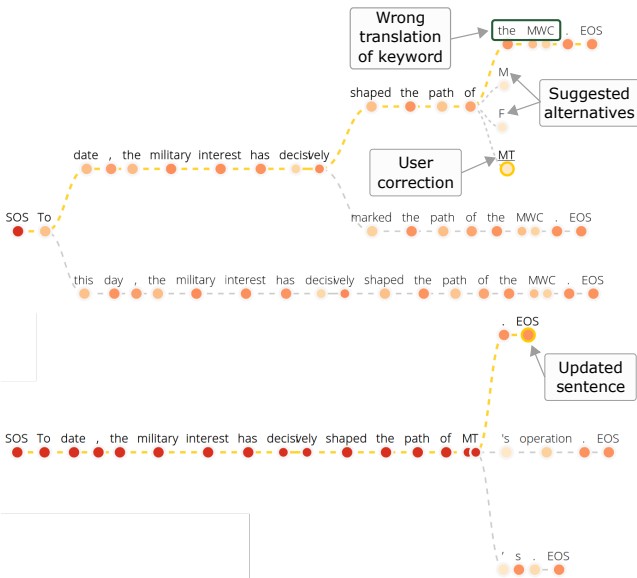

Figure 7: Example of a mistranslated sentence containing the keyphrase 'MÜ' shown as beam search visualization: (top) suggested translation, suggested alternatives, and custom correction; (bottom) updated translation tree for corrected keyword with new suggestions for continuing the sentence after the custom change.

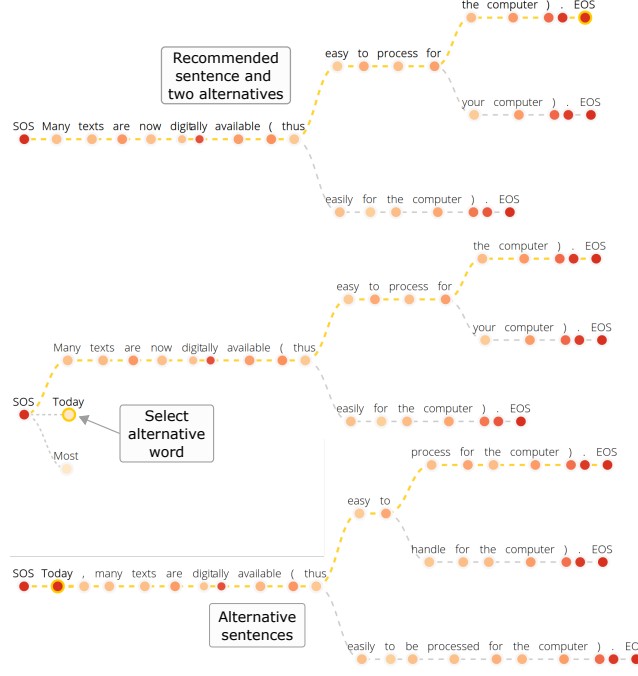

Figure 8: Correctly translated sentence is changed to another correct translation. 'SOS' is selected to show alternative beginnings for the sentence. After choosing an alternative the remaining sentence gets updated by another correct translation.

While this part of the translation is missing, the translated sentence is still correct and fluid; it might be difficult to detect such an error without such attention visualizations.

An example of a wrong translation containing a **keyphrase** is visualized in Figure 7. Here, it is also shown that using the beam search visualization, it is possible to select an alternative translation interactively starting from the position where the first error occurs. The beam search provides possible alternative translations, but it is possible to manually type what the user believes should be the next term. Here, we enter the correct translation manually. The beam search visualization automatically updates in real-time according to the correction.

Finally, it is also possible to change sentences without mistakes. Sometimes, sentences are correctly translated, but different words or sentence structures are used as the current user would prefer for the context of a sentence or to express someone's own style (Figure 8). Again, it is possible to explore and select alternative words or sentences with the Beam Search View. If we wanted to start the sentence with a different word, an alternative could be selected, and the remaining sentence would get updated accordingly.

After correcting and accepting multiple translation corrections, the Document View shows how a translation was changed (Figure 9).

### 4.3 Model Fine-tuning and Retranslation

After users corrected multiple sentences, they can choose to re-train the current model for not yet accepted sentences (R5). The model is then fine-tuned using the corrected sentences by the user. Afterward, the system translates the uncorrected sentences to improve translation quality. Since our document contains 29 times the keyphrase 'MÜ' that is wrongly translated, we retrained our model after correcting only a few (less than 5) of these terms to 'MT'. After retranslation, the Document View shows the difference of the translations compared to before. For both the LSTM and the Transformer model, all or almost all occurrences of 'MÜ' are now correctly translated. The user can look at the changes and accept translations or continue with iteratively improving sentences and fine-tuning the model.

### 4.4 Architecture-specific Observations

We initially designed our approach for the use with an LSTM-based model with an attention mechanism. Since other architectures exist to translate documents, we also adapted it and tested its usefulness

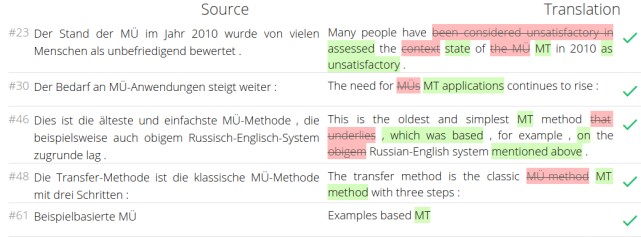

Figure 9: Document View showing corrected translations and changes to the initial machine-generated translations.

Table 2: Ratings from our user study for each evaluated view on a 7-point Likert scale; mean and standard deviation values are provided.

| View | Effectiveness | Visualization | Interaction |
|---|---|---|---|
| **Metrics View** | 5.9 (1.1) | 6.8 (0.4) | 6.1 (0.7) |
| **Keyphrase View** | 4.4 (1.6) | 6.5 (1.2) | 6.3 (1.1) |
| **Beam Search View** | 5.6 (1.5) | 6 (1.3) | 4.5 (1.8) |
| **Attention View** | 5.6 (0.8) | 6.2 (1.2) | 5.9 (0.9) |

for the current state-of-the-art Transformer architecture [42] (R6). This architecture is also attention-based, and we analyzed how well it fits our interactive visualization approach. The general workflow of our system can be used in the same way as the model we initially developed it for: the Document and Metrics Views can be used to identify sentences for further investigation, and sentences can be updated using the Sentence and Beam Search View. The main difference between the Transformer model concerning our approach is the attention mechanism that influences the Attention View and some calculated metric values.

The Transformer architecture uses multiple layers with multiple self-attention heads instead of just attentions between encoder and decoder. There are approaches for the visualization of this more complex attention mechanism [43, 44]. The attention values for Transformers could, for example, show different linguistic characteristics for different attention heads [8]. However, including this into our system would make our approach more complex and not useful for end-users (R7) with little knowledge about this architecture. As a simple workaround to apply our visualization, we discard the self-attention and only use the decoder attention. We explored the influence of decoder attention values from different layers, averaged across all attention heads. Similar to Rikters et al. [32], we noticed that averaging attention from all layers is not meaningful since almost all source words are connected to all target words. Using one of the first layers showed similar results. For the final layer, a better alignment could be seen; however, the last token of the source word received too much attention compared to other words. Instead, using the second last layer showed a similar alignment between source and target words as it is available for the LSTM model. Therefore, we adopt this as a compromise for the use in our Attention View and for calculation of metric values.

Since there are different approaches and architectures developed for NMT, we could incorporate them as well (R6). Some might provide better support in gaining insights into the model and offer different visualization and interaction capabilities. For others, new ways for visualization will have to be investigated.

## 5  USER STUDY

We conducted an early user study during the development of our approach to evaluate our system's concept. We used a prototype with an LSTM translation model. The system had the same views as described before but limited features. A group of visualization and machine learning experts were invited to test our system online for general aspects related to visualization, interaction, and usefulness. Our goal was to make sure that we considered aspects relevant from both the visualization and the machine translation perspective in our system and to improve our approach. The user study was questionnaire-based to evaluate the effectiveness of the system, understandability of visualizations, and usability of interaction techniques. A 7-point Likert scale was used. In this study, the German Wikipedia article for *autonomous driving* (*Autonomes Fahren*) [46] was available to all participants. This allowed the participants to

explore the phenomena we showed previously. The participants claimed to have good English (mean = 5.1, std. dev. = 0.8) and very good German (mean = 6.2, std. dev. = 1.7) knowledge. While the visualization experts claimed to have rather low knowledge about machine learning (mean: 2.5), the machine learning experts similarly indicated lower knowledge for visualization (mean: 3).

First, participants were introduced to the system with a short overview of the features. Then, they could explore the system freely with no time restriction. Afterward, they were asked to participate in a survey regarding the usefulness of our system and its design choices. Additionally, there were free-text sections for further feedback. We recruited 11 voluntary participants from our university (six experts on visualization and five for language processing).

The general effectiveness of translating a large document containing more than 100 sentences with our approach was rated high (mean = 5.6, std. dev. = 1.0) compared to a small document containing up to 20 sentences (mean = 4.5, std. dev. = 1.6). The results for effectiveness, ease of understanding and intuitiveness of visualizations, and ease of interaction are given in Table 2. The ratings for the visualizations were high for all views. Best rated was the Metrics View that additionally had the lowest standard deviation. As not all our user study participants were visualization experts, we noticed that non-experts could also manage to understand and work with parallel coordinate plots. We conclude that our design choice for the visualization of metrics was appropriate. The ratings for interaction were also very high, but there was more variation. Especially the interaction for beam search was rated comparatively low and had the highest standard deviation; two language processing participants ranked it very low (1 and 2) and two (one from each participant group) very high (7). This variation might be the result of the learning curve being different for different participant groups. Since we conducted the user study, we have also improved the interaction in this view. For effectiveness, the Keyphrase View had the lowest rating. We believe the reason is that participants were not able to detect enough mistranslated sentences with this view. However, this might be due to our document provided and may differ for other documents containing more domain-specific vocabulary as we showed in our case study.

In addition, we asked users for general feedback on our approach. Especially the Metrics View received positive feedback. Participants mentioned that it is useful for quickly detecting mistranslations through brushing and linking. For the Beam Search View, one participant noted that the alternatives provided would speed up the correction of translations. For one participant, the Attention View was useful in showing the differences in the sentence structure of different languages. Negative feedback was mostly related to interaction and specific features; some participants suggested new features. Multiple participants noted that the exploration and correction of long sentences are challenging in the Beam Search View as the size of the viewport is limited. Furthermore, a feature to delete individual words and functionality for freezing areas was suggested. From the remaining feedback, we already included, for example, an undo function for the sentence views. Also, to find sentences that might contain similar errors, one participant recommended showing sentences similar to a selected sentence, and we added a respective

metric. Additionally, it was mentioned that confidence scores could be shown in the document list next to each sentence and not only in the Metrics View. This would be helpful to quickly examine the confidence value even if the document is sorted by a different metric (e.g., document order); small histograms were added next to each sentence as a quick quality overview.

## 6 DISCUSSION AND FUTURE WORK

To conclude, we present a visual analytics approach for exploring, understanding, and correcting translations created by NMT. Our approach supports users in translating large domain-specific documents with interactive visualizations in different views, and it allows sentence correction in real-time and model adaption.

Our qualitative user study results showed that our visual analytics system was rated positively regarding effectiveness, interpretability of visualizations, and ease of interaction. The participants mastered the translation process well with our selected visualizations. Especially, our choice of parallel coordinate plots for visualization of multiple metrics and the related interaction techniques for brushing and linking were rated positively. Our approach had a clear preference for translating large documents compared to a traditional text-based approach. Right now, users have to use metrics to decide with which sentence they will start correcting the translations. More research has to be done for better automatic detection of mistranslated sentences. For example, an additional machine learning model could be trained with sentences that were already identified as wrong translations. Additionally, a future step should include a more in-depth user study including our target group. For example, we could evaluate the performance of translation tasks by comparing our interactive approach with a manual method.

We believe that our system is useful for people who have to deal with large documents and could use the features of interactive sentence correction and domain adaption. Comparing the use of our approach for LSTM and the Transfomer architecture showed almost no difference; for both, we could successfully interactively improve the translation quality of documents and see model-specific information. We argue that our general translation and visualization process can also be used with further models, while in such cases, some visualization views might need limited adaptation.

### ACKNOWLEDGMENTS

This work was funded by the Deutsche Forschungsgemeinschaft (DFG, German Research Foundation) under Germany's Excellence Strategy – EXC-2075 – 390740016.

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
