# OpenReview forum: "Visual-Interactive Neural Machine Translation"
_graphicsinterface.org/Graphics_Interface/2021/Conference/Second_Cycle — GI 2021_

### Official Review · Reviewer_yJBu · 2021-04-18
**Good research worthy of publication, but perhaps not ideal for GI**

**Rating:** 6
**Confidence:** 2

**Review:**

This paper presents a visualization tool to analyze and correct machine learning translations between two languages (German and English). Although I have no experience with machine learning or formal (manual) translation methods, the related work section seems adequately thorough. Similarly, the paper details the design of the visual analytics system, and even includes a brief case study of its use. The Abstract is a little bit verbose, but does summarize the research and the results.

However, I do take issue with the user study conducted and described in this paper. The user study does not perform a comparative evaluation. It is comprised of users giving ratings on an ordinal scale. A more appropriate methodology would be to give participants a translation task that they complete using the visual analytics approach, and also using a traditional method. Compare the results with strictly machine-learning translations, according to completion time and accuracy. Try to quantify the amount of improvement from using the visual analytics approach and give statistical results (e.g., ANOVA).

I feel that this research is excellent, and paper is good. However, the content of the paper focuses too much on the translation models, and too little on the user study component. Including a user study that follows an experimental methodology and analysis would transform this conference paper into a journal article. I feel that this work deserves publication, but I am doubtful that GI is the appropriate venue (perhaps UIST?).

---

### Official Review · Reviewer_Yk1n · 2021-04-23
**Good Visualization Design Study Paper**

**Rating:** 7
**Confidence:** 4

**Review:**

In this design study, the authors walk through the design of a visual analytics system for inspecting and correcting translation results from neural models. The paper is well written and easy to follow. The paper also clearly explains the problem domain and the visualization designs, and validates the system with a case study and a user study,

A few references are missing:

Spence Green, Jeffrey Heer, and Christopher D. Manning. 2013. The Efficacy of Human Post-Editing for Language Translation

Spence Green, Jason Chuang, Jeffrey Heer, and Christopher D. Manning. 2014. Predictive Translation Memory: A Mixed-Initiative System for Human Language Translation

Spence Green, Jeffrey Heer, and Christopher D. Manning. 2015. Natural Language Translation at the Intersection of AI and HCI. Communications of the ACM

The above references look at the problem of supporting translators to make sense and correct mistakes made by machine translation. It’ll be useful to have a discussion in the paper on how the proposed approach relates to and differs from these related papers.

Some questions that can be clarified in the revision:

“Keyphrases are domain- specific words and were not often included in the training data used for our model”: why is it so?

Sentence view: “After adding a new word in the text area, the translation with the highest score is used for the remainder of the sentence”, what happens if user deletes or changes a word?

Model fine tuning: how many user corrections are necessary to have an impact on model performance, can you quantify the impact of user corrections on model performance?

---

### Official Review · Reviewer_7ciB · 2021-05-05
**Integrated visual analytics/retraining loop for MT**

**Rating:** 8
**Confidence:** 4

**Review:**

This paper presents a multi-layer visualization for machine translation. Views include a parallel coordinates plot (metrics view), a list of keyphrases, a view of the document, and tools for examining the details of the translation (attention view and beam search view).

The paper is well-written and clear. The approach builds on past research in understanding the machine translation search space and diagnosing/correcting errors in MT. The related work is sufficiently described. I thought the paper was well-framed in the requirements list, which guide the design principles (though, these requirements were not clearly grounded in the real-world tasks and needs of the target audience of translators). The approach to design is based on a drill-down visualization, from the document level to a target sentence, then allowing for exploration of translation alternatives and correction. The real novelty of this work comes from the feedback loop in which edits to the translations, either through point-and-click adjustments in the beam search view or through manual edits, are fed back to adjust the rest of the translations in the document. I think this type of approach could be very useful in targeting a general MT model to a specific domain.  The various search and filtering metrics would clearly be useful at targeting sentences to correct.

I think the biggest weakness of this work is in the user study which was only briefly described. I appreciate that experts from both visualization and machine learning were used, however, neither of these groups were from the previously stated target user group of translators (R7). I think this selection of participants leads to conclusions which will not generalize. For example, it is claimed that parallel coordinates plots worked surprisingly well for the participants, but all participants were technically trained computer scientists. PCPs have been shown in the past to be difficult to use by a general audience. I also thought there was a missed opportunity to verify whether through using the tools the translation actually improved (or, the model improved). That is, a third party could review the translation before and after editing through the tool, or the accuracy of the translation model could be assessed after retraining through the document editing process.

Despite these suggestions for improvement, I think this paper presents enough novel material, in particular the tightly integrated feedback loop, and is clearly written and presented, so it should be accepted.

---

### Meta-Review · Area_Chair_wkbW · 2021-05-05

**Recommendation:** Accept
**Confidence:** 4

**Metareview:**

All the reviewers agree that the research presented in this paper is strong and the writing is clear. Considering the novelty and soundness of research contributions, this paper should be accepted at GI. One of the weaknesses of the paper mentioned by multiple reviewers is the user study. The authors may consider conducting a more in-depth user study and addressing the concerns raised in the reviews.

---

### Decision · Program_Chairs · 2021-05-08

Accept